Late Holocene spread of pastoralism coincides
with endemic megafaunal extinction on
Madagascar. *Proc. R. Soc. B* **288**: 20211204.

palaeontology, environmental science

extinction, megafauna, pastoralism,
palaeoecology, radiocarbon, competition

**Author for correspondence:**
Sean W. Hixon
e-mail: hixon@ucsb.edu

Electronic supplementary material is available
online at https://doi.org/10.6084/m9.figshare.
c.5494700.

# Late Holocene spread of pastoralism coincides with endemic megafaunal extinction on Madagascar

Sean W. Hixon[1], Kristina G. Douglass[2], Brooke E. Crowley[3,4],
Lucien Marie Aimé Rakotozafy[5], Geoffrey Clark[6], Atholl Anderson[6],
Simon Haberle[6,7], Jean Freddy Ranaivoarisoa[8], Michael Buckley[9],
Salomon Fidiarisoa[10], Balzac Mbola[10] and Douglas J. Kennett[1]

[1]Department of Anthropology, University of California at Santa Barbara, Santa Barbara, CA, USA
[2]Department of Anthropology, Pennsylvania State University, State College, PA, USA
[3]Department of Geology, and [4]Department of Anthropology, University of Cincinnati, Cincinnati, OH, USA
[5]Institute of Civilizations, Museum of Art and Archaeology, University of Antananarivo, Antananarivo, Madagascar
[6]Department of Archaeology and Natural History, College of Asia and the Pacific, and [7]Australian Research
Council Centre of Excellence for Australian Biodiversity and Heritage, School of Culture, History and Language
(CAP), The Australian National University, Canberra, Australian Capital Territory, Australia
[8]Department of Biological Anthropology and Sustainable Development, University of Antananarivo, Madagascar
[9]School of Natural Sciences, Manchester Institute of Biotechnology, The University of Manchester, Manchester
M1 7DN, UK
[10]Département des Sciences Biologiques, Faculté des Sciences, Université de Tuléar, BP 185, Tuléar, 601,
Madagascar

SWH, 0000-0001-6147-7118; KGD, 0000-0003-0931-3428; BEC, 0000-0002-8462-6806;
MB, 0000-0002-4166-8213

Recently expanded estimates for when humans arrived on Madagascar (up to
approximately 10 000 years ago) highlight questions about the causes of the
island's relatively late megafaunal extinctions (approximately 2000–500
years ago). Introduced domesticated animals could have contributed to extinc-
tions, but the arrival times and past diets of exotic animals are poorly known.
To conduct the first explicit test of the potential for competition between intro-
duced livestock and extinct endemic megafauna in southern and western
Madagascar, we generated new radiocarbon and stable carbon and nitrogen
isotope data from the bone collagen of introduced ungulates (zebu cattle,
ovicaprids and bushpigs, *n* = 66) and endemic megafauna (pygmy hippo-
potamuses, giant tortoises and elephant birds, *n* = 68), and combined these
data with existing data from endemic megafauna (*n* = 282, including giant
lemurs). Radiocarbon dates confirm that introduced and endemic herbivores
briefly overlapped chronologically in this region between 1000 and 800 cali-
brated years before present (cal BP). Moreover, stable isotope data suggest
that goats, tortoises and hippos had broadly similar diets or exploited similar
habitats. These data support the potential for both direct and indirect forms
of competition between introduced and endemic herbivores. We argue that
competition with introduced herbivores, mediated by opportunistic hunting
by humans and exacerbated by environmental change, contributed to the
late extinction of endemic megafauna on Madagascar.

## 1. Introduction

Until quite recently, Madagascar's diverse endemic fauna included gorilla-sized
lemurs (*Archaeoindris fontoynontii*), giant tortoises (*Aldabrachelys* spp.), three-
metre-tall elephant birds (*Aepyornis maximus*) and one-metre-tall pygmy
hippos (*Hippopotamus* spp.). However, all of the island's endemic animals
greater than 10 kg are now extinct, and introduced zebu cattle (*Bos taurus indi-
cus*) are currently the largest animal on the island. A wide range of potential

Proc. R. Soc. B 288: 20211204

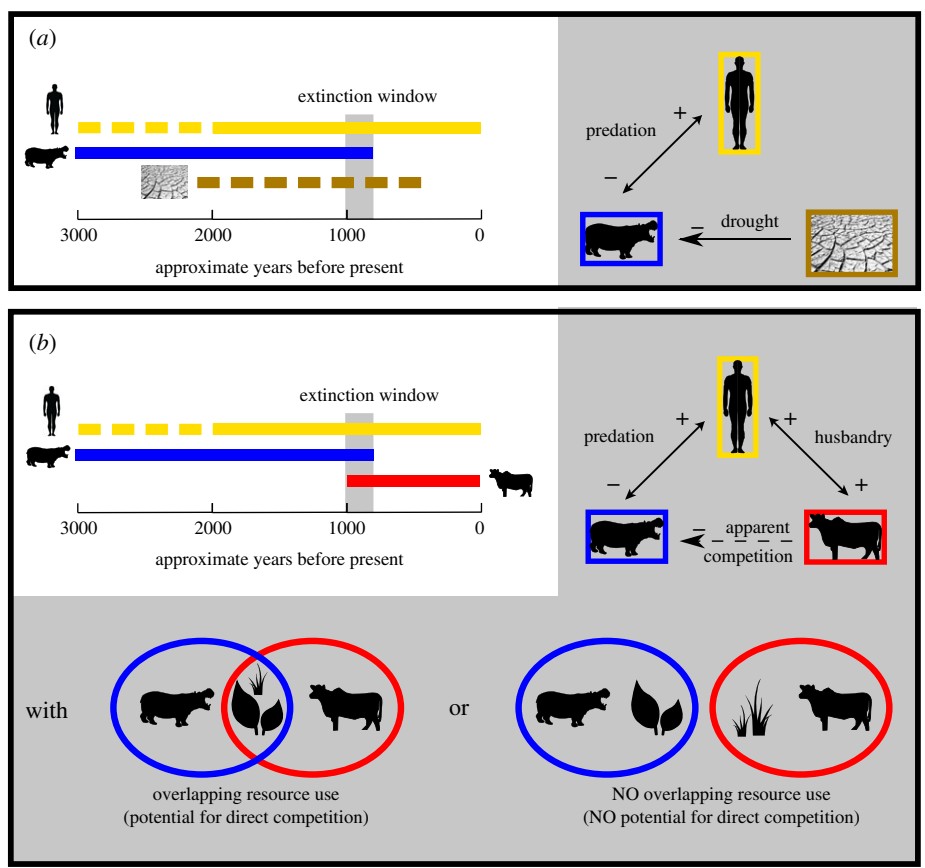

**Figure 1.** Two possible scenarios for megafaunal extinction, with human arrival and megafaunal extinction estimates drawn from [14]C reviews [5,6]. Dashed lines represent uncertain timing of human introduction and aridification, blue represents extinct endemic megafauna and red represents introduced herbivores. The 'synergy' hypothesis involves impacts of drought and overhunting (a), and the 'subsistence shift' hypothesis involves apparent competition and possibly direct competition between introduced livestock and endemic megafauna (b). These versions of the 'synergy' and 'subsistence shift' hypotheses are not mutually exclusive: negative synergistic effects involving drought and introduced livestock could have contributed to megafaunal extinction. (Online version in colour.)

stressors, including overhunting, drought, fire, disease and biological invasion, could have contributed to extinctions in Madagascar [1]. We integrate radiocarbon dating with stable isotope analysis of introduced ungulate and extinct endemic megaherbivore bone collagen to test extinction hypotheses that involve competition between endemic and introduced herbivores.

A debate regarding when humans first arrived in Madagascar is ongoing. Some researchers favour early human arrival 10 000–4000 years ago based on rare stone tools and cutmarks on ancient elephant bird bone [2,3], while others favour recent arrival 1600–1000 years ago based on broader cultural considerations [4]. Here, we assume that people were on the island 2000–1600 years ago based on a recent review of radiocarbon ([14]C) data associated with traces of human activity [5]. However, it was not until approximately 1000 years ago that most populations of endemic megafauna crashed [6]. These extinctions coincided with drought in parts of southern Madagascar [7,8]. Still, the idea that island-wide aridification drove extinctions is inconsistent with (i) the persistence of diverse Malagasy megafauna during relatively severe Pleistocene climate fluctuations, (ii) megafaunal bone stable isotope records that suggest few directional changes in habitat aridity [9,10] and (iii) palaeoclimate records that reveal asynchronous changes in regional climate during the late Holocene [8,11]. While Madagascar's late Holocene fossil record is rich, the early Holocene and particularly Pleistocene records are unfortunately sparse

and limited to less than 20 [14]C-dated individuals from central and northern Madagascar [6].

An alternative hypothesis is that the spread of pastoralism and farming approximately 1000 years ago triggered megafaunal extinction [12,13]. Palaeoenvironmental records document vegetation change in multiple regions of Madagascar during the past millennium [14,15], but there is a dearth of data from directly [14]C-dated introduced animals [5]. In southern and western Madagascar (hereafter referred to as southwest Madagascar), the foddering needs of livestock currently influence the movement of pastoralists, and people modify both grasslands and forests through selective clearance and the propagation of introduced succulents to sustain livestock [16].

We consider potential for interactions among humans, livestock and endemic megafauna that may be direct (e.g. resource competition or hunting) or indirect (e.g. mediated by hunting/predation, as in apparent competition). Direct competition between introduced and endemic animals resulting from overlap in diet or habitat use can be partially inferred through stable isotope analysis of consumer tissue. Evidence for hunting comes from butchery marks on Malagasy megafaunal bone [2,4]. Hunting could have acted in concert with regional aridification to drive megafaunal extinction (an early formulation of the 'synergy hypothesis' [17]; figure 1), or it could have occurred with pastoralism to create apparent competition between introduced and endemic herbivores ('subsistence shift hypothesis' [12]; figure 1). Apparent competition occurs between two species that are prey for the same predator/hunter [18], and it

disadvantages prey that are relatively sensitive to predation (e.g. slowly reproducing megafauna that are not part of animal husbandry). By definition, apparent competition does not depend on overlapping resource use of potential competitors (as inferred from stable isotope and other data). Instead, support for apparent competition comes from evidence of (i) contemporaneous shared predation/hunting pressure (as inferred through cutmarked and [14]C-dated bone) and (ii) different impacts on prey populations (assumed in our case due to differences in animal husbandry versus bushmeat hunting). Regardless of direct competition among herbivores, livestock could have devastated endemic megafauna by facilitating human population growth and overhunting.

A precise chronology of pastoralism and species extinctions on Madagascar is needed to test the 'subsistence shift' hypothesis. Lack of temporal overlap between introduced and endemic megaherbivores is inconsistent with the latter hypothesis (figure 1). Here, we directly [14]C date exotic ungulates from southwest Madagascar to test the possibility that biological invasion contributed to a pulse of megafaunal extinction. We report data from ovicaprids (*Capra hircus* and *Ovis aries*), bushpigs (*Potamochoerus larvatus*) and zebu (*Bos taurus indicus*). We focus on the relatively abundant and well-preserved bone deposits of southwest Madagascar. Based on reports of mixed megafaunal and introduced herbivore bones [19], we predict that the arrival of livestock preceded megafaunal extinction.

Direct competition can follow from niche overlap, and stable carbon isotope ($\delta^{13}$C) and nitrogen isotope ($\delta^{15}$N) values of bone collagen give a unique insight into the past niches (diet and habitat) of individuals and groups. The photosynthetic pathway of plants strongly influences their $\delta^{13}$C values and those of consumer tissue [20]: plants that use the $C_3$ photosynthetic pathway (primarily trees, shrubs and herbs) tend to be depleted in $^{13}$C relative to plants that use the $C_4$ pathway (primarily grasses) or CAM pathway (primarily succulents). Relatively open soil nitrogen cycling in arid environments drives relatively high ecosystem $\delta^{15}$N values [21]. To a lesser extent, local environmental conditions, including canopy cover and coastal proximity, can also influence $\delta^{13}$C and $\delta^{15}$N values [22]. Existing data suggest that endemic megaherbivores were browsers [23], so we predict that they had overlapping $\delta^{15}$N and $\delta^{13}$C values with browsing goats, and not with introduced grazers such as zebu and sheep. While similar isotope values may reflect overlaps in resource use that create direct competition (figure 1), direct competition does not necessarily follow from overlaps in resource use. This is because (i) animals may partition resources by foraging on different plants, at different times, or in different areas with similar isotope values [24], and (ii) the impacts of herbivory on plant communities are diverse [25]. By contrast, non-overlapping $\delta^{15}$N and $\delta^{13}$C values suggest distinct diets and habitat use and leave little potential for direct competition (figure 1).

## 2. Methods

### (a) Study area/regional overview

Southwest Madagascar experiences a prolonged dry season and receives only a brief rainy season during the austral winter. Regional vegetation is dominated by deciduous $C_3$ trees and CAM succulents, as well as some $C_4$ grasses [26]. Riparian forests dissect this otherwise dry landscape, but relatively high $\delta^{15}$N values in subfossil lemur collagen suggest that these animals

did not prefer wet corridors [27]. The similar environmental conditions across southwest Madagascar justify the comparison of stable isotope data from plants and animals across the region [22]. To make relatively fined-grained comparisons of stable isotope data, we also used two approaches to group data from multiple sites: (i) five ecogeographic site groups defined by coasts and drainages with comparable aridity (figure 2) and (ii) simple inland versus coastal groups (coastal defined as less than 10 km from shore).

### (b) Data collection

Details regarding all aspects of sample selection, laboratory analysis, data review and data analysis are provided in the electronic supplementary material, appendix. We sampled skeletal remains from 66 introduced zebu, sheep, goats and bushpigs as well as 71 endemic tortoises, hippos and elephant birds from 21 sites in southwest Madagascar (figure 2; electronic supplementary material, appendix datasets S1 and S2). We extracted and purified bone collagen at the Pennsylvania State University (PSU), gathered stable carbon and nitrogen isotope data at Yale University's W. M. Keck Biotechnology Resource Laboratory, and submitted 111 ancient samples with sufficient preservation for analysis at the PSU AMS [14]C Laboratory or the UC Irvine W. M. Keck Carbon Cycle AMS Laboratory (electronic supplementary material, appendix 'Sample Collection' and 'Laboratory Analyses'). We co-analysed these new data with previously published regional [14]C data from 155 megafaunal bones and eggshells, as well as stable isotope data from 261 specimens belonging to five endemic taxa (electronic supplementary material, appendix 'Additional Data'). The published dataset includes data from two extinct lemur taxa: the giant ruffed lemur (*Pachylemur insignis*) and monkey lemur (*Archaeolemur majori*). The monkey lemur had a semi-terrestrial locomotor strategy [28] that would have made it more likely to interact with introduced ungulates, and a relatively large number of giant ruffed lemur bones have been directly [14]C-dated. We ultimately excluded 39 specimens from the analysis of combined data due to data quality issues. The final co-analysed dataset includes reliable [14]C dates ($n = 238$), $\delta^{13}$C values ($n = 374$) and $\delta^{15}$N values ($n = 293$) from 45 sites (figure 2).

### (c) Data analysis

We estimated introduction and extinction times based on sequences of [14]C data using both classical and Bayesian approaches to control for differences in sample size (figure 3; electronic supplementary material, appendix table S1). General linear models (electronic supplementary material, appendix 'Data Analysis', tables S2 and S3) compared influences of taxon, space and time on stable isotope values. We fitted Bayesian ellipses to stable isotope data for each taxon to visualize overlap in isotopic niche space (figure 4; electronic supplementary material, appendix figure S6 and table S4). Stable isotope mixing models informed by modern plant $\delta^{13}$C data (electronic supplementary material, appendix dataset S3 and figures S2–S3) expanded dietary inference (electronic supplementary material, appendix figure S4). To establish end-members for our mixing model, we combined published data from the region ($n = 492$) with new $\delta^{13}$C data for 242 plant samples collected at three sites in the vicinity of Tulear/Toliara, southwest Madagascar and analysed at the University of Cincinnati (electronic supplementary material, appendix dataset S3).

## 3. Results

### (a) Chronological overlap

Radiocarbon-dated bone and eggshell suggest comparable introduction times among herbivores and a brief overlap

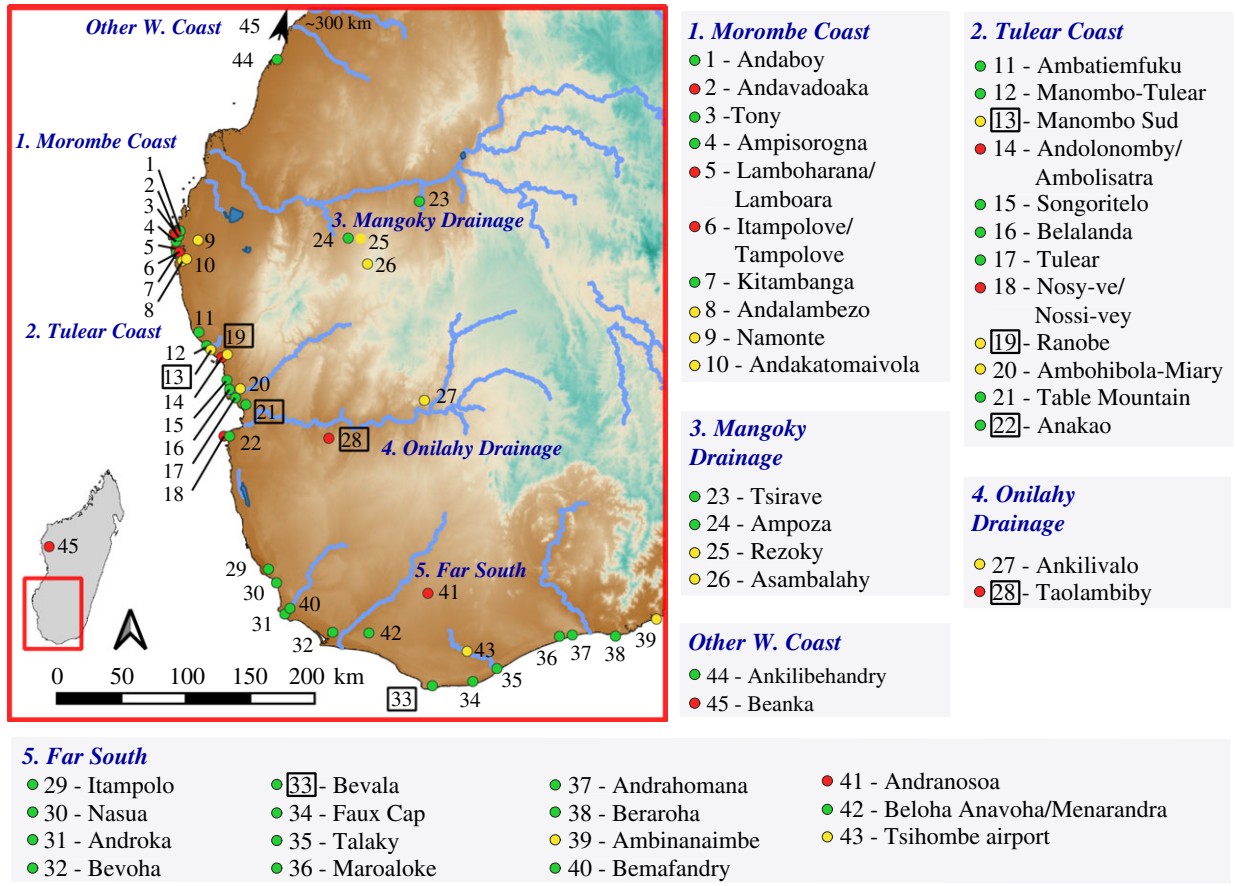

**Figure 2.** Map of 45 study sites in southwest Madagascar included in this study. Sites marked in green include bones of only endemic animals, those in yellow include only introduced animal bone, and those in red include both. We used sites grouped along the Morombe coast [1–10], Tulear coast [11–22], Mangoky drainage [23–26], and Onilahy drainage [27,28] and sometimes the Far South [29–43] for comparisons of stable isotope values among taxa (electronic supplementary material, appendix 'Data Analysis – Radiocarbon & Stable Isotopes'). Specimens from two other west coast sites [44,45] contribute ¹⁴C data to our analyses but are not considered in comparisons of stable isotope data given their relatively wide geographical spread. Plant stable isotope data come from specimens collected in the vicinity of sites outlined in black. (Online version in colour.)

between introduced and extinct endemic megafauna in southwest Madagascar. Reliable $^{14}$C dates from endemic megafaunal bone and eggshell ($n = 186$) span the Holocene, yet approximately 95% of the specimens are younger than 4000 calibrated years before present (cal BP). All the $^{14}$C-dated introduced animals ($n = 52$) are younger than 1020 cal BP. The earliest $^{14}$C-dated introduced animals are a sheep from Andranosoa (figure 2, PSUAMS-8684, 1165 ± 25 $^{14}$C BP, 1060–960 cal BP) and a contemporaneous zebu from the same site (PSUAMS-8685, 1150 ± 15 $^{14}$C BP, 1060–960 cal BP). These individuals pre-date the last known individuals of the giant ruffed lemur *P. insignis* (from Tsirave, CAMS-167930, 940 ± 20 $^{14}$C BP, 900–740 cal BP) and hippo (from Lamboharana/Lamboara, PSUAMS-5629, 1100 ± 15 $^{14}$C BP, 980–930 cal BP). Both the earliest $^{14}$C-dated bushpig (PSUAMS-5619, 975 ± 15 $^{14}$C BP, 910–790 cal BP) and what is possibly the earliest $^{14}$C-dated goat (an ovicaprid tentatively identified as *Capra hircus*, PSUAMS-3764, 900 ± 20 $^{14}$C BP, 800–730) come from Andavadoaka, and these animals also likely overlapped temporally with the last known *P. insignis* in the region. The last securely dated giant tortoise (from Lamboharana/Lamboara, PSUAMS-5131, 1155 ± 15 $^{14}$C BP, 1060–960 cal BP) and elephant bird (from Ambolisatra/Andolonomby, OxA-33535, 1237 ± 24 $^{14}$C BP, 1180–1000 cal BP) likely overlapped temporally with the zebu and sheep at Andranosoa and pre-date the earliest introduced bushpigs and goats by possibly less than 100 years.

Classical and Bayesian 95% confidence/credible intervals suggest that all regional extinctions and introductions occurred over the course of less than 500 years between 1200 and 700 cal BP (figure 3; electronic supplementary material, appendix 'Data Analysis – Radiocarbon', and table S1). During this interval, individuals of all nine endemic and introduced taxa could have briefly co-occurred and interacted. The Bayesian and classical approaches suggest that the maximum temporal overlap between introduced and endemic herbivores was approximately 500–600 years (shared between zebu and *P. insignis*).

## (b) Niche overlap

Herbivore $\delta^{13}$C and $\delta^{15}$N values demonstrate various degrees of potential niche overlap among hippos, giant tortoises, both giant lemur species (*A. majori* and *P. insignis*) and ovicaprids (figure 4). For example, *A. majori* and *P. insignis* share approximately 80% of their isotopic niche space with goats, yet this overlap accounts for less than 45% of the isotopic niche space occupied by goats. Isotopic overlap between introduced and endemic herbivores appears to have been greater at coastal sites ($n = 144$ individuals from 17 sites), yet this could be a product of relatively limited sampling at inland sites ($n = 139$ individuals from 9 sites, electronic supplementary material, appendix figure S6).

Six general linear models based on multiple approaches to grouping data according to collection site suggest that taxon, age, location and interactions among all variables typically best explain variability in faunal $\delta^{13}$C and $\delta^{15}$N values

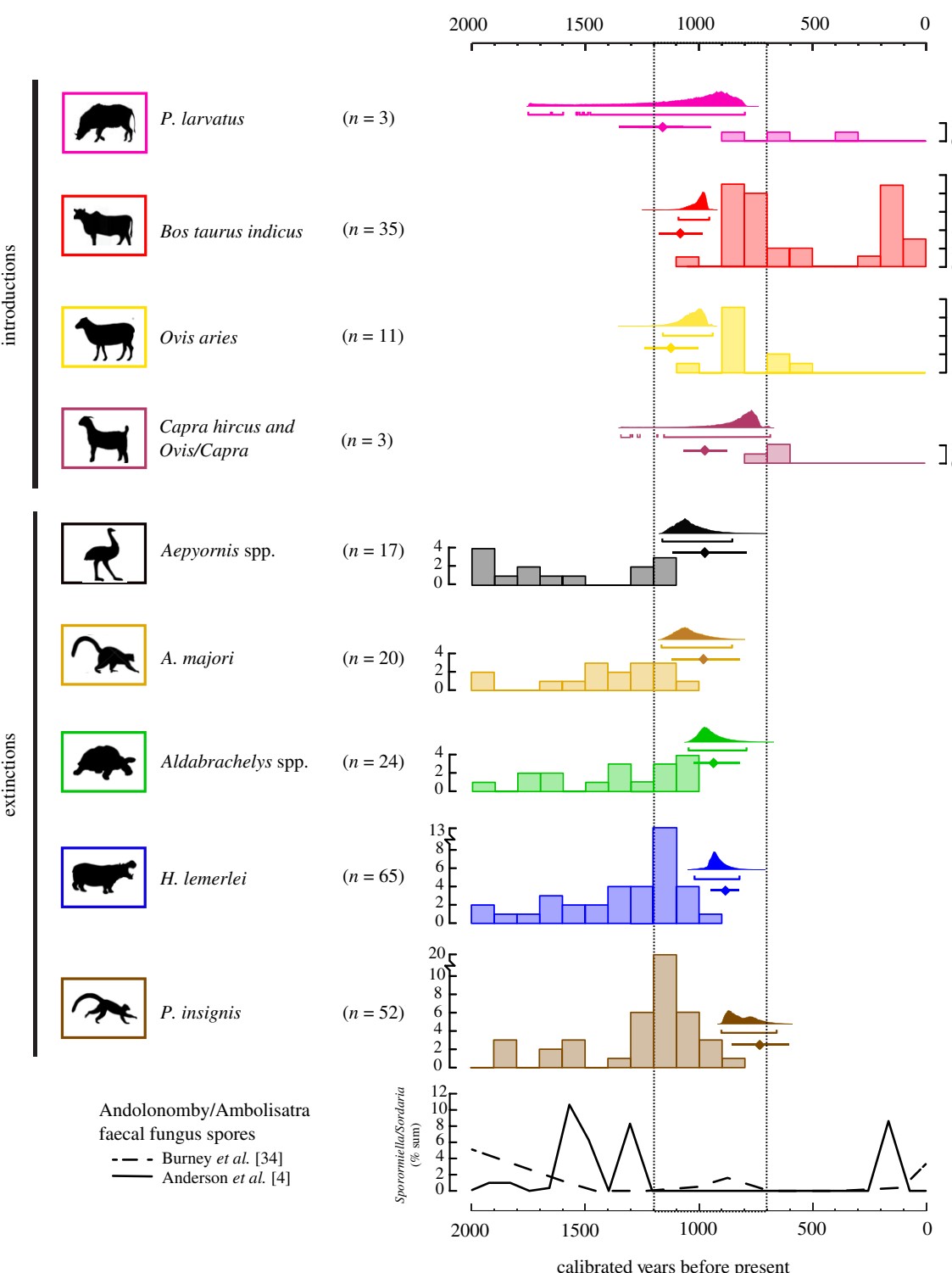

**Figure 3.** Taxon-specific [14]C date histograms (unsaturated colour) and confidence/credible intervals (saturated colour) for extinction and introduction event estimation. Bayesian posterior probability distributions with 95% brackets give the probability that a species is not present at a certain time given that it was not sampled. Lines with median estimates marked as diamonds give the 95% confidence intervals produced from the application of a classical frequentist approach to the problem of event estimation. Estimates from the classical approach give the probability that a species was not sampled at a certain time given that it was not present. Note that two [14]C-dated unspecified *Ovis/Capra* are approximately contemporary with a [14]C-dated *Capra hircus* (PSUAMS-3693), and these three specimens are grouped to create confidence/credible intervals for goat introduction. The dashed line highlights a 500-year period that probably included the depicted series of introductions and extinctions. Published records of faecal fungus spores in Andolonomby sediment may reflect local megaherbivore abundance and are included for comparison [4,34]. (Online version in colour.)

(electronic supplementary material, appendix table S2 and S3, and see electronic supplementary material, appendix 'Data Analysis – Stable Isotopes' for a full discussion of models). These models highlight differences in $\delta^{13}$C values between coastal and inland individuals, and these differences create greater similarity in $\delta^{13}$C values among herbivores at coastal sites.

Because there are no widespread and consistent effects of site group or coastal proximity on $\delta^{13}$C or $\delta^{15}$N values across herbivore taxa, we feel secure in considering isotopic niche

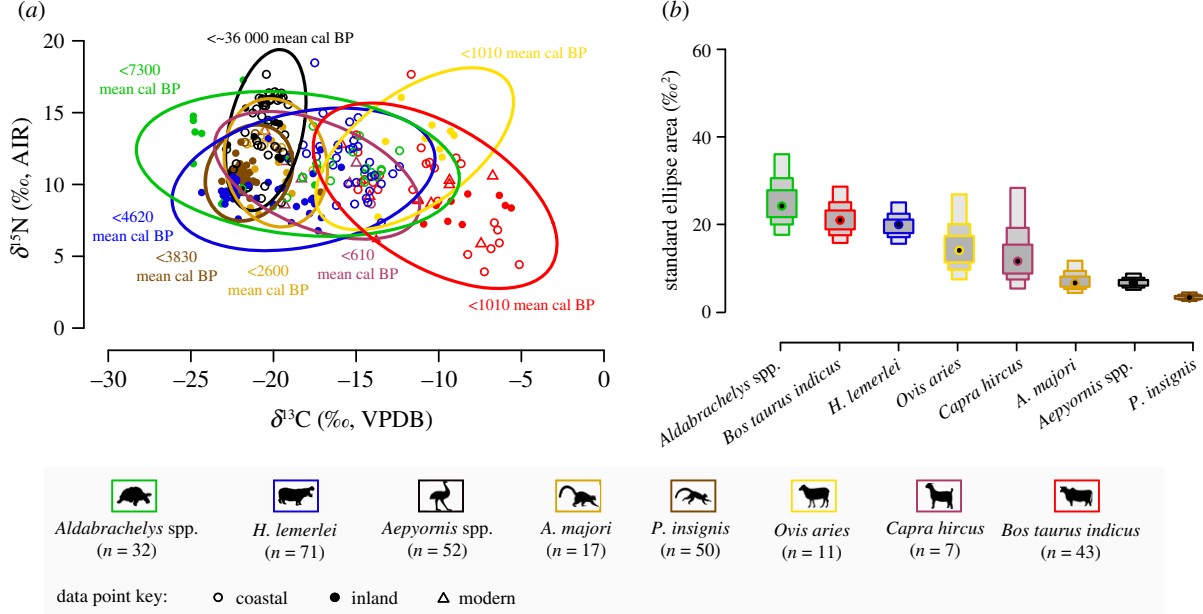

**Figure 4.** Taxon-specific stable isotope values in bone collagen space ((*a*), *n* = 283), with bolded ellipses that outline approximately 95% of the data from each group. All data are from site groups 1–5. Solid points represent individuals from inland sites, and hollow points represent individuals from coastal sites. Modern individuals of *Capra hircus* and *Bos taurus indicus* are marked as triangles. Stable carbon isotope data from *Aepyornis* spp. eggshell organics are corrected so that they are comparable to bone collagen values (electronic supplementary material, appendix 'Data Analysis –Stable Isotopes'). Only one *Aepyornis* spp. individual on this figure is represented by bone collagen data (UCIAMS-224190, $\delta^{13}C = -22.7‰$, $\delta^{15}N = 10.7‰$). Mean calibrated date ranges next to ellipses specify the range of ages for samples in each group. Sample size corrected standard ellipse areas (SEAc, containing approximately 40% of the data from each group, (*b*)) give relative estimates for the similarity of stable isotope values between individuals of the same taxon. SEAc estimates for each taxonomic group appear as modes (points) surrounded by shaded boxes that span 50%, 75%, and 95% of the estimates (from dark to light). See electronic supplementary material, appendix figure S6 for ellipses fitted to coastal and inland data separately. (Online version in colour.)

overlap among taxa from all site groups. The relatively large areas of isotopic niche space occupied by goats, sheep, zebu, hippos and giant tortoises (sample size corrected standard ellipse areas [SEAc] ≥ 15‰$^2$, figure 4) suggest that these animals all exploited more varied resources and habitats than elephant birds or giant lemurs (SEAc ≤ 10‰$^2$). To some extent, SEAc values are sensitive to the amount of time and space (i.e. number of sites) integrated in each group (electronic supplementary material, appendix table S4). However, predictor variables of temporal spread (defined as the period spanning 50% of the mean calibrated dates) and number of sites sampled within each taxonomic group cannot fully account for the variance in SEAc values (electronic supplementary material, appendix 'Data Analysis – Stable Isotopes'). The inability of temporal and geographic spread to explain variation in SEAc values suggests that differences in these values correspond to dietary differences among taxa. Goats and sheep are noteworthy, because they have one of the shortest temporal spreads (≤200 years) and come from one of the smallest number of sites (*n* = 2 for sheep, *n* = 4 for goats), yet each occupies a relatively large isotopic niche space (SEAc ≥ 15‰$^2$). By contrast, elephant bird samples also come from a similarly limited number of sites (*n* = 3, including two inland sites) yet form a relatively small isotopic niche (SEAc = 6.8‰$^2$) despite having the largest temporal spread (greater than 10 000 years).

Plant $\delta^{13}C$ values document expected patterns among photosynthetic groups, with significantly lower values in $C_3$ plants (*n* = 537, $\bar{x} = -27.1‰$) than in CAM succulents (*n* = 136, $\bar{x} = -13.3‰$) or $C_4$ grasses (*n* = 61, $\bar{x} = -12.3‰$; electronic supplementary material, dataset S3, and see appendix 'Data Analysis – Stable Isotopes' for additional discussion of plant $\delta^{13}C$ data). Large $\delta^{13}C$ ranges for zebu (*n* = 45, 10.7‰, from

−15.9‰ to −5.2‰) and ovicaprids (*n* = 20, 11.9‰, from −20.5‰ to −8.6‰) are consistent with modern observations that these animals consume a diversity of grasses, shrubs and endemic succulents [29,30]. Elevated $\delta^{13}C$ values for zebu and sheep suggest that these animals generally consumed more CAM or $C_4$ plants (succulents and grasses) than did other introduced livestock and endemic megafauna. Indeed, the results of a mixing model involving all $\delta^{13}C$ values from modern plants in the region suggest that CAM or $C_4$ plants comprised approximately 90% of zebu diet and approximately 80% of sheep diet on average (electronic supplementary material, appendix 'Data Analysis – Stable Isotopes', and figure S4). All other herbivores (including goats and possibly introduced bushpigs) likely consumed more $C_3$ plant material than CAM and $C_4$ plant material. Of the extinct megaherbivores, hippos consumed the most CAM or $C_4$ plant material (approx. 35% of hippo diet).

For those taxa with sufficient sample sizes and temporal spreads (zebu, hippos and giant tortoises), we can observe site group-specific changes through time in collagen stable isotope values (figure 5; electronic supplementary material, appendix figure S5). There are no significant increases in megafaunal $\delta^{13}C$ values over time, despite pollen records from the coastal site of Andolonomby/Ambolisatra (site group 2) that document the decline of arboreal plant taxa over the last 2000 years (figure 5*c*) [14]. As forests declined, hippo $\delta^{13}C$ values apparently decreased at coastal site groups 1 and 2 (electronic supplementary material, appendix figure S5A), and giant tortoise $\delta^{13}C$ values significantly decreased at site group 1 (*n* = 9, $r_s$ = 0.66, *p* = 0.05). Meanwhile, zebu $\delta^{13}C$ values apparently increased at site groups 1 and 2 (electronic supplementary material, appendix figure S5A).

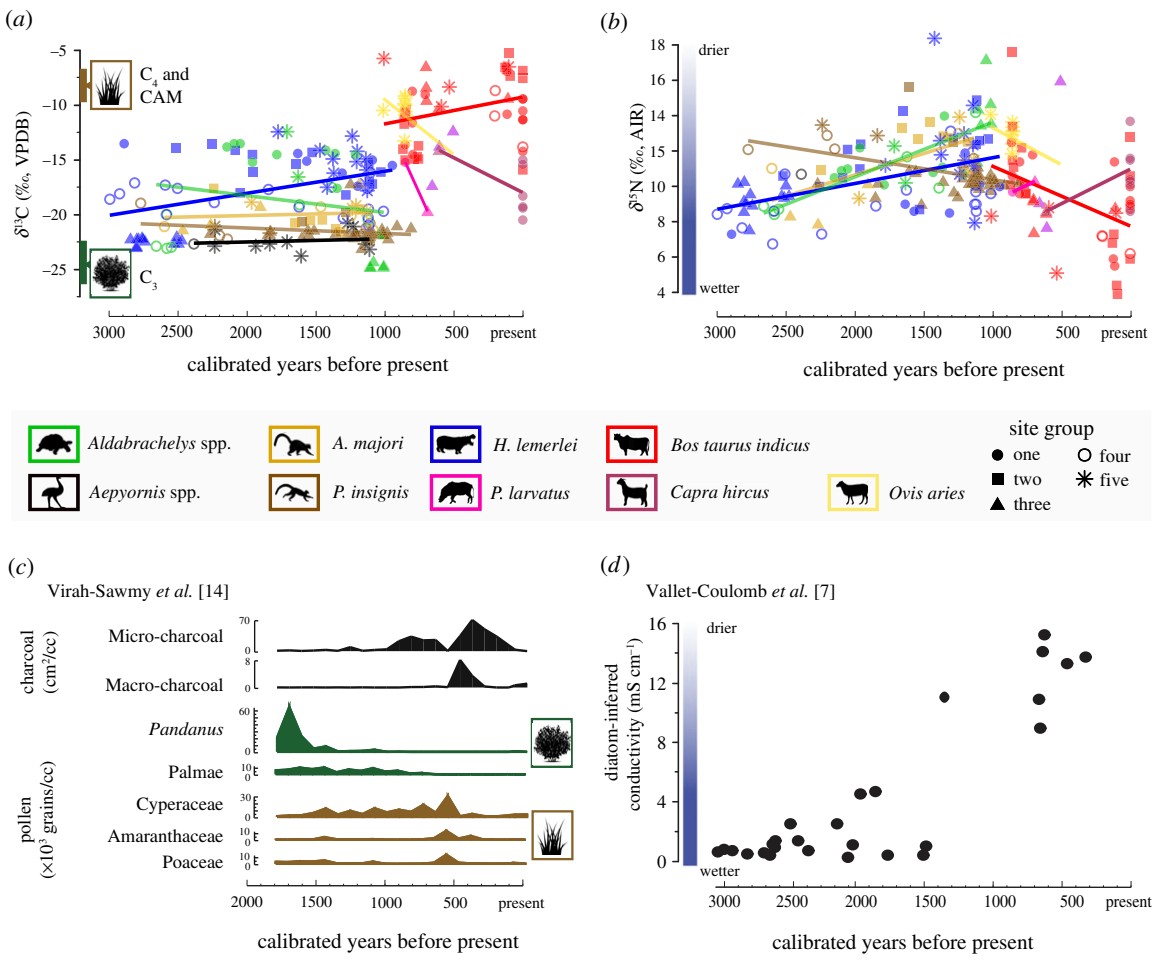

**Figure 5.** Stable (*a*) carbon (*n* = 228) and (*b*) nitrogen (*n* = 212) isotope values through time, with points coloured according to taxon (following figures 3 and 4), points shaped according to site group (following figure 2), regression lines for all data from each taxon, and published regional records of vegetation change (*c*) and aridity (*d*) for comparison [7,14]. C$_3$ and C$_4$ & CAM plant $\delta^{13}$C ranges are based on modern data from southwest Madagascar in collagen space. Note that uneven sampling across sites through time can explain some of the region-wide changes in taxon-specific isotope records. See electronic supplementary material, appendix figure S5 for temporal trends in $\delta^{13}$C and $\delta^{15}$N values plotted separately and according to both site group and taxon for taxa with relatively large sample sizes and temporal spreads. (Online version in colour.)

Megafauna $\delta^{15}$N values and a contemporary record of the salinization of Lac Ihotry (near site group 1, figure 5*b*,*d* [7]) suggest that at least the coastal animals may have tracked local aridification. Hippo collagen $\delta^{15}$N values significantly increased at coastal site group 2 (*n* = 15, $r_s = -0.55$, *p* = 0.03) and apparently increased at coastal site group 1 and inland site group 4 (electronic supplementary material, appendix figure S5B). Tortoise collagen $\delta^{15}$N values also apparently increased at site group 1. Despite this drying trend, zebu collagen $\delta^{15}$N values significantly decreased at site groups 1 and 2 (*n* = 15 and 16, respectively, and $r_s \geq 0.54$, $p \leq 0.03$, electronic supplementary material, appendix figure S5B).

## 4. Discussion

Brief temporal overlap between introduced livestock and endemic megafauna is consistent with a key component of the 'subsistence shift' hypothesis for megafaunal extinction [12], and isotopic niche overlap among coastal goats, hippos and giant tortoises indicates that direct forms of competition could have existed in some cases between introduced and endemic herbivores. Regardless of when humans first arrived on the island, multiple lines of evidence suggest

that the spread of pastoralism in southwest Madagascar contributed to megafaunal extinction.

Radiocarbon data suggest that zebu, sheep, goats and bushpigs all became established in southwest Madagascar between 1,200 and 700 years ago, but $^{14}$C datasets from goats and bushpigs are still limited. This timing coincided with significant growth and movement of Malagasy populations [31], the rise of the island's earliest urban centre [32], and the expansion of trade along the west coast of the island [33]. Here we have shown that species introductions also coincided with the regional extinction of five endemic herbivore taxa. The co-occurrence of livestock and endemic megafauna that we document was possibly brief and is not visible in records of large herbivore faecal fungus spores from lake sediments [34]. However, the chronological overlap is consistent with mixed archaeological/palaeontological deposits and historical accounts of endemic megafauna [19].

During the brief co-occurrence of livestock and endemic megafauna, pastoralists both relied on their livestock and hunted endemic animals [4,35]. Predation impacts populations of prey species to different extents, and this likely put introduced and endemic herbivores into an apparent competition that negatively affected endemic species (figure 1). The impact of human predation on livestock populations was minimal. Pastoralists kill livestock, and ovicaprids are sometimes

considered pests because of their diverse and voracious appetites, yet animal husbandry typically gives a net benefit to both humans and domesticates. This is particularly true for zebu, which are currently repositories of wealth for pastoralists in southwest Madagascar and are used in a variety of ceremonies and rites. Pastoralists expand their zebu herds by protecting them from predators, modifying vegetation to create a reliable supply of fodder and moving to track suitable habitat [16,36]. Zebu outnumbered people on Madagascar by as much as 2 : 1 in the early twentieth century [16], and approximately 55% of Madagascar's surface was dedicated to pastoralism in 2000 [37]. The expansion of zebu populations likely contributed to human population growth, which indirectly facilitated the hunting of endemic animals. Endemic herbivores likely did not benefit from animal husbandry and became increasingly susceptible to predation as humans transformed forests for livestock. Small-bodied lemurs, such as the endemic sifaka *P. verreauxi*, have sustained hunting pressure from humans for at least a millennium, and this pressure may have caused reductions in body size [38]. Extinct endemic herbivores with long lifespans, relatively slow reproductive rates, and few ways to escape terrestrial predators (e.g. giant tortoises) were also butchered [4,35] and probably suffered the greatest from the increase in hunting pressure that livestock husbandry facilitated [39].

Isotopic niche overlap among goats, hippos and giant tortoises indicates that there was some potential for direct forms of competition between introduced and endemic herbivores. Elephant birds probably had a relatively distinct isotopic niche from introduced herbivores, yet our results suggest that overlaps in collagen stable isotope values among taxa varied both temporally and spatially, so identifying niche overlap at a particular time and site group remains challenging. Still, existing data suggest that isotopic niche overlap may have been greater at coastal than inland sites. Future research should compare the vegetation histories of coastal and inland southwest Madagascar and expand faunal datasets to identify any lags between coastal human settlement and inland extinctions. Particular attention should be given to goats, which are notorious invaders of other island ecosystems that can browse some endemic plants to near extinction [40]. Perhaps the most extreme impact of goats on vegetation and other herbivores comes from the Galápagos, where goats are known to decimate vegetation during the dry season and leave little browse for endemic giant tortoises [41]. In Madagascar, people often encourage ovicaprids to forage on a wide range of plant types in deference to the grazing preferred by zebu. The dispersal ability of giant tortoises may attest to the high tolerance of resource depression [42], but our findings underscore that the potential for direct competition must be considered during ongoing efforts to both reintroduce giant tortoises and maintain local livelihoods [30,43]. Future recovery and analysis of additional ovicaprid bones should be a priority, because even our limited dataset follows from sampling all known collections of ovicaprid bones from the region.

Vegetation change and aridification likely affected interactions among humans and herbivores. For the following reasons, our isotopic data suggest that vegetation change was likely the primary stress for endemic populations while aridification could have been a greater stress for introduced livestock. Forest cover declined in coastal southwest Madagascar during the last 2000 years [8,14], yet $C_4$ grasses and CAM succulents remained a minor component of hippo and tortoise diet before their extinction, and carbon isotope data suggest that elephant birds and giant lemurs (*A. majori* and *P. insignis*) may have relied exclusively on C3 vegetation. Dwindling patches of forest left less browse for endemic megafauna but cleared land for the vegetation that introduced herbivores generally prefer. Forest clearance during this time was widespread and may have been anthropogenic; similar transitions to a grassy biome occurred in the absence of drought in northern and central Madagascar [15,44]. At the same time, increases in collagen $\delta^{15}$N values of hippos and possibly other taxa indicate that these animals survived aridification in southwest Madagascar by persisting in increasingly arid habitat. Meanwhile, relatively low $\delta^{15}$N values in zebu collagen suggest that they may have tracked relatively moist habitat. Modern Malagasy herders move their animals in pursuit of freshwater and succulent fodder during the dry season [36]. In this light, consistent past reliance on relatively moist habitat by ancient zebu may reflect their sensitivity to drought, which is an ongoing concern in southern Madagascar.

Understanding past changes in herbivory in Madagascar is important for conservation efforts. In our existing sample, we observe that introduced ungulates are unique in their isotopic niche breadth and reliance on $C_4$ or CAM vegetation. Despite the mixed woodlands and $C_4$-dominated grasslands that currently exist in many regions on the island, our expanded $\delta^{13}$C data are consistent with previous work that suggests Madagascar lacked an endemic grazer guild [23]. Extinctions of large herbivores can have cascading negative effects on remaining plant and animal species due to the various ways in which large herbivores consume vegetation, redistribute resources and modify the physical environment [45]. For example, experimental data suggest that the Malagasy giant tortoises aided the dispersal and germination of baobab seeds [46], and extant hippos from Africa play important roles in both cycling nutrients [47] and maintaining the structural heterogeneity of riparian vegetation [48]. Likewise, introduced bushpigs on Madagascar may help disperse large seeds, but the full extent to which introduced species and extant endemic species have continued the ecosystem services of now-extinct megafauna is unknown. Introduced ruminants are poor candidates for facilitating large seed dispersal, particularly given that zebu had limited dietary overlap with endemic megafauna.

Combined stable isotope and radiocarbon data give us unique insight into late Holocene pulses of biodiversity loss in southwest Madagascar and contribute to a growing body of evidence that neither climate change nor hunting pressure alone consistently drive extinction (e.g. [49,50]). In southwest Madagascar, direct and human-mediated indirect interactions between introduced and endemic herbivores, which involved an increase in human population, likely contributed to megafaunal extinction to a greater extent than novel hunting pressure or regional aridification. This pattern is not unique to Madagascar. Indeed, shifts in human subsistence also may help explain lags between human presence and extinction elsewhere. For example, relatively late changes in stone tool technologies and patterns of human subsistence may help explain a prolonged period of coexistence (30 ka) between humans and megafauna in the Indian subcontinent [51]. Also, in North America, the introduction of a novel hunting pressure alone cannot explain the late Holocene

extinction of California's flightless sea duck that occurred only after a protracted (8 ka) period of coexistence and hunting [52]. However, increasing human reliance on marine resources during the late Holocene associated with the proliferation of sedentary coastal communities might explain the sea duck's late extinction [53]. In Eurasia, Koch & Barnosky [54] suggest that hominins who specialized in pursuing large-bodied prey tracked booms and busts in prey populations on timescales of 100 ka and that it was the arrival and proliferation of anatomically modern humans with broad diets that ultimately contributed to megafaunal extinctions. The similarity in these cases follows from the general observation that a predator population that specializes in one type of prey will track changes in the prey population, while one that relies on a diversity of prey can easily overexploit the prey species that cannot sustain heavy predation [55]. These interactions among hunters and potential competitors must be considered when drawing parallels between recent overkill on islands and earlier extinctions in continental settings [56]. We identify Madagascar as an ideal place to further study this mechanism of extinction due to the potentially early arrival of people, the island's short extinction chronology and the relatively recent arrival of pastoralism.

Data accessibility. All data are provided as electronic supplementary material, and all details regarding data collection and analysis are provided in the appendix. Data also available from the Dryad Digital Repository: https://doi.org/10.5061/dryad.41ns1rndq [57].

The data are provided in the electronic supplementary material [58].

Authors' Contributions. S.W.H.: conceptualization, data curation, formal analysis, funding acquisition, investigation, methodology, project administration, resources, visualization and writing-original draft; K.G.D.: conceptualization, funding acquisition, investigation, project administration, resources, writing-review and editing; B.E.C.: conceptualization, formal analysis, funding acquisition, methodology, resources, writing-review and editing; L.M.A.R.: conceptualization, investigation, resources, writing-review & editing; G.C.: data curation, funding acquisition, investigation, resources, writing-review and editing; A.A.: data curation, investigation, resources, writing-review and editing; S.H.: investigation, resources, writing-review and editing; J.F.R.: investigation, methodology, resources, writing-review and editing; M.B.: formal analysis, funding acquisition, investigation, methodology, resources, writing-review and editing; S.F.: investigation, methodology and resources; B.M.: investigation, methodology and resources; D.J.K.: conceptualization, funding acquisition, investigation, methodology, project administration, resources, supervision, writing-original draft, writing-review and editing

All authors gave final approval for publication and agreed to be held accountable for the work performed therein.

Competing interests. We declare we have no competing interests

Funding. This research was supported by National Science Foundation grants GRFP 2015213455 (S.W.H.), Archaeology DDRI 1838393 (D.J.K. and S.W.H.) and BCS 1749676 (B.E.C.), and by additional funding from the Royal Society UF120473 (M.B.), Sigma Xi, the American Philosophical Society, Society for Archaeological Science, the Max Planck Institute for the Science of Human History, PSU Energy and Environmental Sustainability Laboratories, PSU Africana Research Center and PSU Anthropology Department. The NSF Archaeometry Program (BCS-1460367 (D.J.K.)) and the Pennsylvania State University (D.J.K.) provided general laboratory support.

Acknowledgements. We thank P. Brewer, G. Billet, C. Argot, B. Kear, S. Goodman, H. Wright, H. Randrianatoandro, O. Griffiths and D. Burney for help with sampling bones from existing collections, the crew of the Morombe Archaeological Project, the communities of the Velondriake Marine Protected Area, B Manjakahery and D. Damy for their assistance during field collection, D. Burney and L. Godfrey for comments on the manuscript, and L. Eccles, B. Culleton, R. Wood and J. Southon for assistance during laboratory analysis.

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
