## [Peer Review File · Proceedings of the Royal Society B: Biological Sciences]

Review History

RSPB-2021-0573.R0 (Original submission)

Review form: Reviewer 1

Recommendation

Major revision is needed (please make suggestions in comments)

Scientific importance: Is the manuscript an original and important contribution to its field?

Acceptable

General interest: Is the paper of sufficient general interest?

Good

Quality of the paper: Is the overall quality of the paper suitable?

Acceptable

Is the length of the paper justified?

Yes

Should the paper be seen by a specialist statistical reviewer?

Yes

Do you have any concerns about statistical analyses in this paper? If so, please specify them explicitly in your report.

Yes

It is a condition of publication that authors make their supporting data, code and materials available - either as supplementary material or hosted in an external repository. Please rate, if applicable, the supporting data on the following criteria.

Is it accessible?

Yes

Is it clear?

Yes

Is it adequate?

Yes

Do you have any ethical concerns with this paper?

No

Comments to the Author

I thought that this paper was interesting and well-written.

I would like to see a few points addressed or explained in the text before publication:

- 1) a more explicit stance on the arrival time of humans and domesticated fauna in Madagascar -- this matters for the interpretation of overlapping timelines and impact on megafaunal extinction.
- 2) More discussion of sample size limitations -- I am especially concerned by the small sample sizes for goats and sheep as these are given the most emphasis in the results and discussion terms of niche overlap and potential contribution to megafaunal extinctions.
- 3) I am also concerned that the bayesian inference of temporal overlap with 14C for goats put (n=3) put them at the tail end/outside of potential overlap with extinct species.

Review form: Reviewer 2

Recommendation

Major revision is needed (please make suggestions in comments)

Scientific importance: Is the manuscript an original and important contribution to its field?

Good

General interest: Is the paper of sufficient general interest?

Good

Quality of the paper: Is the overall quality of the paper suitable?

Acceptable

Is the length of the paper justified?

Yes

Should the paper be seen by a specialist statistical reviewer?

No

Do you have any concerns about statistical analyses in this paper? If so, please specify them explicitly in your report.

No

It is a condition of publication that authors make their supporting data, code and materials available - either as supplementary material or hosted in an external repository. Please rate, if applicable, the supporting data on the following criteria.

Is it accessible?

Yes

Is it clear?

Yes

Is it adequate?

Yes

Do you have any ethical concerns with this paper?

No

Comments to the Author

- Line 51 and elsewhere: why refer to zebu as *Bos taurus/indicus*? Better to just decide on one scientific name?

- Lines 54-56: this sentence should be the last sentence of the introduction, not at end of first paragraph.

- Lines 61-63: I think you need to provide much more robust justification as to why you discount the older evidence of human presence on Madagascar. Just because early records (stone tools etc) are rare, this doesn't mean that they're not valid evidence of humans; this seems very fallacious. Surely you need to offer more justification? Rarity could reflect many things, e.g. limited sampling of older Holocene horizons, or low density early human settlers... Also, given the focus of the paper, do you even need to be stating anything about your assumptions on timing of first colonisation? What you're interested in here is the arrival/expansion of pastoralism, which can clearly happen at a different time to first human arrival in an island system, given the potential for multiple waves of human colonisation, inter-regional trade and cultural exchange, etc.

- Lines 69-71: See my previous point - here you admit that the older record is still sparse, thus making it possible that this is the reason for limited evidence of early human presence. Also, I'm not sure what point is being made here, by saying that older records are limited? (i.e. is it relevant to the wider topics being considered here?)

- Line 89: surely the subsistence shift hypothesis refers to people changing their diet from megafauna to smaller species, i.e. shifting their subsistence strategy - not to evidence of competition between megafauna and domestic animals? I think you need a different name for this hypothesis?

- Line 89 onwards: I don't think that you need to even introduce the concept of "apparent competition" here - to me, it's rather confusing and potentially misleading, especially when referred to alongside other types of competition. You can address the issues/hypothesis here easily without needing to refer to it in this way; to me it just confuses the point you're trying to make. Apparent competition can be difficult to identify in nature, often because of the complexity of indirect interactions that involve multiple species and changing environmental conditions, and I'm not sure that harvesting domesticated cattle versus hunting wild megafauna necessarily

represents a good/real example of apparent competition.

- Paragraph starting at line 100: how much temporal overlap would you expect to see though? (especially given the known incompleteness of the fossil/archaeological record, although I appreciate that you use statistical techniques to account for this.) Also -- what would be inconsistent with the synergy hypothesis? I think you should state this too. Basically, what would disprove either hypothesis, within the context of known issues with the quality of the Holocene record?

- The introduction ends really abruptly. You need to end the introduction with a clear statement of what you actually want to do in this study... so put the last sentence from the first intro paragraph here?

- Line 157: why just investigate these two extinct lemurs? What other species were there in the SW region? Does excluding these others introduce any possible biases in interpreting extinction drivers? Also - are these widely-used common names for these extinct species? (and you refer to one of them variably as "monkey lemur" and "giant monkey lemur", which is confusing.)

- Line 159, "we ultimately excluded 39 specimens" - are these specimens of lemurs (the way it's written implies this), or of other samples from your overall dataset? Also, the text here is very confusing - why specifically mention lemur species here, but not species diversity in hippos, elephant birds and tortoises? - both in terms of which species you sampled, and the total species diversity of these groups in the SW region. It's very hard to assess the implications of your data when you don't even state clearly what species you actually studied, and how much of the region's wider megafaunal diversity they actually represent. You need to state here the number of samples you analysed per megafaunal species. Were all taxa ID'd to species level or genus level? If not to species level, is this likely to matter? (e.g. is there a possibility for species-level differences in dietary ecology within faunal groups, e.g. as a mechanism to support sympatric species diversity?)

- Line 169: it's unclear what you were using these GLMs specifically to investigate - need to say more than just that you used them to compare values. What were your responses and predictors, and what was the question you were asking by using the models?

- Line 205: also need to state minimum temporal overlaps between species, and for how many species.

- Discussion, opening paragraph: Here onwards, you basically talk solely about the (in my opinion mistakenly named) subsistence shift hypothesis. What about the aridification/synergy hypothesis? You need to also explain what your results say about that, and to be testing between these two hypotheses using your data, since you set them both up as competing hypotheses in the introduction.

- How do your results relate chronologically with the known period of forest clearance in Madagascar c.1000 years ago? i.e. if you don't factor that in too, how can you tease out the relative effects of competition versus habitat loss?

- Lines 297-8 ("and this likely put introduced and endemic herbivores into apparent competition that negatively affected endemic species"), lines 307-8 ("which indirectly facilitated the hunting of endemic animals"), lines 329-330 ("people may have encouraged ovicaprids to forage on a wide range of plant types"): these statements all feel like unsupported assumptions.

- Line 322: more niche overlap at coastal sites - can you gain any insights from considering possible vegetation assemblage structural/diversity differences in coastal versus inland sites? I assume there will be some? Also, how exactly is "coastal" defined - how far from actual coast? Is

there a distinct coastal vegetation belt/ecoregion?

- Line 330-331, "The tolerance of Malagasy giant tortoises to resource depression could have been high": again, this statement feels very speculative without further explanation in the text. Overall, a lot of the discussion feels rather speculative.

- Line 337, "while aridification could have been a greater stress for introduced livestock": how do your results demonstrate this?

- Line 343, "Forest clearance during this time was widespread and may have been anthropogenic": pretty sure it was definitely anthropogenic!? What evidence do you have that it wasn't? See my earlier comments about needing to consider the potential relative effects of introduced mammals and human-caused habitat loss alongside each other, rather than focusing almost entirely on the effects of potential competition.

- Line 346, "tracked aridification": I think you're using this term incorrectly – at least in terms of how it's used in the ecological literature. To track aridification, surely an animal would move from one landscape to another to maintain similar levels of aridification, rather than staying put and experiencing a change in aridification. That's how ecologists refer to species tracking changing climatic conditions, etc.

- Paragraph starting line 353: but, did you sample all extinct megafaunal taxa? If not, then you can't make broad statements like this - as unsampled species may have filled different isotopic/trophic niches. Indeed, it also feels like much of this paragraph doesn't really relate to your specific results.

- Line 372: again, what about habitat loss? That could have been the KEY extinction driver in this system – the spread of introduced mammals could just have been secondary to this. This is a really essential point that currently feels almost completely missing from the discussion and how you've interpreted your data.

- Figures – why do you refer to the elephant birds as "Aepyornis spp."? The only elephant bird you mention specifically in the main text is Vorombe titan, and referring to these birds as "Aepyornis" is technically excluding both Vorombe and Mullerornis. What species of elephant birds did you actually sample?

Decision letter (RSPB-2021-0573.R0)

21-Apr-2021

Dear Mr Hixon:

I am writing to inform you that your manuscript RSPB-2021-0573 entitled "Late Holocene spread of pastoralism coincides with endemic megafaunal extinction on Madagascar" has, in its current form, been rejected for publication in Proceedings B.

This action has been taken on the advice of referees, who have recommended that substantial revisions are necessary. With this in mind we would be happy to consider a resubmission, provided the comments of the referees are fully addressed. However please note that this is not a provisional acceptance.

The resubmission will be treated as a new manuscript. However, we will approach the same reviewers if they are available and it is deemed appropriate to do so by the Editor. Please note

that resubmissions must be submitted within six months of the date of this email. In exceptional circumstances, extensions may be possible if agreed with the Editorial Office. Manuscripts submitted after this date will be automatically rejected.

Please ensure you address the important point raised about the small sample size.

Sincerely,
Dr John Hutchinson, Editor
mailto: proceedingsb@royalsociety.org

Associate Editor
Board Member: 1
Comments to Author:
Dear Authors,

Thank you for your submission to PRSB. Both reviewers recognise your work as a generally well written and potentially interesting contribution, though they both identify similar concerns with the manuscript in its present form.

Most importantly, both reviewers raise important questions about interpretations of human arrival times on Madagascar, as well as the small sample sizes for certain domesticated species and the extent to which those small sample sizes may introduce interpretive ambiguities.

Reviewer 2 further raises questions about how best to integrate discussions of human arrival times in the manuscript. Critically, Reviewer 2 goes on to request substantially more detail regarding the specimens investigated in this study and their taxonomic assignments, as well as making numerous additional comments that should be addressed.

Reviewer(s)' Comments to Author:
Referee: 1

Comments to the Author(s)
I thought that this paper was interesting and well-written.

I would like to see a few points addressed or explained in the text before publication:

- 1) a more explicit stance on the arrival time of humans and domesticated fauna in Madagascar -- this matters for the interpretation of overlapping timelines and impact on megafaunal extinction.

2) More discussion of sample size limitations -- I am especially concerned by the small sample sizes for goats and sheep as these are given the most emphasis in the results and discussion terms of niche overlap and potential contribution to megafaunal extinctions.

3) I am also concerned that the bayesian inference of temporal overlap with 14C for goats put (n=3) put them at the tail end/outside of potential overlap with extinct species.

Referee: 2

Comments to the Author(s)

- Line 51 and elsewhere: why refer to zebu as *Bos taurus/indicus*? Better to just decide on one scientific name?

- Lines 54-56: this sentence should be the last sentence of the introduction, not at end of first paragraph.

- Lines 61-63: I think you need to provide much more robust justification as to why you discount the older evidence of human presence on Madagascar. Just because early records (stone tools etc) are rare, this doesn't mean that they're not valid evidence of humans; this seems very fallacious. Surely you need to offer more justification? Rarity could reflect many things, e.g. limited sampling of older Holocene horizons, or low density early human settlers... Also, given the focus of the paper, do you even need to be stating anything about your assumptions on timing of first colonisation? What you're interested in here is the arrival/expansion of pastoralism, which can clearly happen at a different time to first human arrival in an island system, given the potential for multiple waves of human colonisation, inter-regional trade and cultural exchange, etc.

- Lines 69-71: See my previous point - here you admit that the older record is still sparse, thus making it possible that this is the reason for limited evidence of early human presence. Also, I'm not sure what point is being made here, by saying that older records are limited? (i.e. is it relevant to the wider topics being considered here?)

- Line 89: surely the subsistence shift hypothesis refers to people changing their diet from megafauna to smaller species, i.e. shifting their subsistence strategy - not to evidence of competition between megafauna and domestic animals? I think you need a different name for this hypothesis?

- Line 89 onwards: I don't think that you need to even introduce the concept of "apparent competition" here - to me, it's rather confusing and potentially misleading, especially when referred to alongside other types of competition. You can address the issues/hypothesis here easily without needing to refer to it in this way; to me it just confuses the point you're trying to make. Apparent competition can be difficult to identify in nature, often because of the complexity of indirect interactions that involve multiple species and changing environmental conditions, and I'm not sure that harvesting domesticated cattle versus hunting wild megafauna necessarily represents a good/real example of apparent competition.

- Paragraph starting at line 100: how much temporal overlap would you expect to see though? (especially given the known incompleteness of the fossil/archaeological record, although I appreciate that you use statistical techniques to account for this.) Also -- what would be inconsistent with the synergy hypothesis? I think you should state this too. Basically, what would disprove either hypothesis, within the context of known issues with the quality of the Holocene record?

- The introduction ends really abruptly. You need to end the introduction with a clear statement of what you actually want to do in this study... so put the last sentence from the first intro paragraph here?

- Line 157: why just investigate these two extinct lemurs? What other species were there in the SW region? Does excluding these others introduce any possible biases in interpreting extinction drivers? Also - are these widely-used common names for these extinct species? (and you refer to one of them variably as “monkey lemur” and “giant monkey lemur”, which is confusing.)

- Line 159, “we ultimately excluded 39 specimens” - are these specimens of lemurs (the way it’s written implies this), or of other samples from your overall dataset?
Also, the text here is very confusing - why specifically mention lemur species here, but not species diversity in hippos, elephant birds and tortoises? - both in terms of which species you sampled, and the total species diversity of these groups in the SW region. It’s very hard to assess the implications of your data when you don’t even state clearly what species you actually studied, and how much of the region’s wider megafaunal diversity they actually represent. You need to state here the number of samples you analysed per megafaunal species. Were all taxa ID’d to species level or genus level? If not to species level, is this likely to matter? (e.g. is there a possibility for species-level differences in dietary ecology within faunal groups, e.g. as a mechanism to support sympatric species diversity?)

- Line 169: it’s unclear what you were using these GLMs specifically to investigate - need to say more than just that you used them to compare values. What were your responses and predictors, and what was the question you were asking by using the models?

- Line 205: also need to state minimum temporal overlaps between species, and for how many species.

- Discussion, opening paragraph: Here onwards, you basically talk solely about the (in my opinion mistakenly named) subsistence shift hypothesis. What about the aridification/synergy hypothesis? You need to also explain what your results say about that, and to be testing between these two hypotheses using your data, since you set them both up as competing hypotheses in the introduction.

- How do your results relate chronologically with the known period of forest clearance in Madagascar c.1000 years ago? i.e. if you don’t factor that in too, how can you tease out the relative effects of competition versus habitat loss?

- Lines 297-8 (“and this likely put introduced and endemic herbivores into apparent competition that negatively affected endemic species”), lines 307-8 (“which indirectly facilitated the hunting of endemic animals”), lines 329-330 (“people may have encouraged ovicaprids to forage on a wide range of plant types”): these statements all feel like unsupported assumptions.

- Line 322: more niche overlap at coastal sites - can you gain any insights from considering possible vegetation assemblage structural/diversity differences in coastal versus inland sites? I assume there will be some? Also, how exactly is “coastal” defined - how far from actual coast? Is there a distinct coastal vegetation belt/ecoregion?

- Line 330-331, “The tolerance of Malagasy giant tortoises to resource depression could have been high”: again, this statement feels very speculative without further explanation in the text. Overall, a lot of the discussion feels rather speculative.

- Line 337, “while aridification could have been a greater stress for introduced livestock”: how do your results demonstrate this?

- Line 343, “Forest clearance during this time was widespread and may have been anthropogenic”: pretty sure it was definitely anthropogenic!? What evidence do you have that it wasn’t? See my earlier comments about needing to consider the potential relative effects of introduced mammals and human-caused habitat loss alongside each other, rather than focusing almost entirely on the effects of potential competition.

- Line 346, “tracked aridification”: I think you’re using this term incorrectly – at least in terms of how it’s used in the ecological literature. To track aridification, surely an animal would move from one landscape to another to maintain similar levels of aridification, rather than staying put and experiencing a change in aridification. That's how ecologists refer to species tracking changing climatic conditions, etc.

- Paragraph starting line 353: but, did you sample all extinct megafaunal taxa? If not, then you can’t make broad statements like this - as unsampled species may have filled different isotopic/trophic niches. Indeed, it also feels like much of this paragraph doesn’t really relate to your specific results.

- Line 372: again, what about habitat loss? That could have been the KEY extinction driver in this system – the spread of introduced mammals could just have been secondary to this. This is a really essential point that currently feels almost completely missing from the discussion and how you've interpreted your data.

- Figures – why do you refer to the elephant birds as “Aepyornis spp.”? The only elephant bird you mention specifically in the main text is Vorombe titan, and referring to these birds as “Aepyornis” is technically excluding both Vorombe and Mullerornis. What species of elephant birds did you actually sample?

Author's Response to Decision Letter for (RSPB-2021-0573.R0)

See Appendix A.

RSPB-2021-1204.R0

Review form: Reviewer 3

Recommendation

Accept as is

Scientific importance: Is the manuscript an original and important contribution to its field?

Excellent

General interest: Is the paper of sufficient general interest?

Excellent

Quality of the paper: Is the overall quality of the paper suitable?

Excellent

Is the length of the paper justified?

Yes

Should the paper be seen by a specialist statistical reviewer?

No

Do you have any concerns about statistical analyses in this paper? If so, please specify them explicitly in your report.

No

It is a condition of publication that authors make their supporting data, code and materials available - either as supplementary material or hosted in an external repository. Please rate, if applicable, the supporting data on the following criteria.

Is it accessible?

Yes

Is it clear?

Yes

Is it adequate?

Yes

Do you have any ethical concerns with this paper?

No

Comments to the Author

I did not review the first version of this manuscript, but I have read through the author responses to reviewer comments from the first submission. The author responses seem to be thorough and clearly explained.

I learned much about the zooarcheology of Madagascar from reading this manuscript, and I thought the authors did a nice job establishing the context of their study. Since my expertise lies in bio(paleo)geochemistry, I focused on the presentation and analyses of these data. The authors apply the appropriate quantitative and modeling approaches (chronological and niche overlap) to the isotopic and radiocarbon data, and the results of these analyses support interpretations discussed in the paper. These are robust datasets that significantly expand the SI and 14C data from Madagascar during this time period.

Overall, the current version of this manuscript is well written and clearly presented - I do not have any substantive suggestions for edits or changes. The results are compelling, and fill in an interesting and important piece of the megafaunal extinction puzzle in Madagascar.

Decision letter (RSPB-2021-1204.R0)

22-Jun-2021

Dear Mr Hixon

I am pleased to inform you that your Review manuscript RSPB-2021-1204 entitled "Late Holocene spread of pastoralism coincides with endemic megafaunal extinction on Madagascar" has been accepted for publication in Proceedings B. Congratulations!!

We are in the awkward position that we could not obtain the prior reviewers, but the Associate Editor and a new reviewer agree that the MS has improved tremendously. Well done.

The referee(s) do not recommend any further changes. Therefore, please proof-read your manuscript carefully and upload your final files for publication. Because the schedule for publication is very tight, it is a condition of publication that you submit the revised version of

your manuscript within 7 days. If you do not think you will be able to meet this date please let me know immediately.

To upload your manuscript, log into <http://mc.manuscriptcentral.com/prsb> and enter your Author Centre, where you will find your manuscript title listed under "Manuscripts with Decisions." Under "Actions," click on "Create a Revision." Your manuscript number has been appended to denote a revision.

You will be unable to make your revisions on the originally submitted version of the manuscript. Instead, upload a new version through your Author Centre.

1) A text file of the manuscript (doc, txt, rtf or tex), including the references, tables (including captions) and figure captions. Please remove any tracked changes from the text before submission. PDF files are not an accepted format for the "Main Document".

2) A separate electronic file of each figure (tiff, EPS or print-quality PDF preferred). The format should be produced directly from original creation package, or original software format. Please note that PowerPoint files are not accepted.

3) Electronic supplementary material: this should be contained in a separate file from the main text and the file name should contain the author's name and journal name, e.g. `authorname_procb_ESM_figures.pdf`

All supplementary materials accompanying an accepted article will be treated as in their final form. They will be published alongside the paper on the journal website and posted on the online figshare repository. Files on figshare will be made available approximately one week before the accompanying article so that the supplementary material can be attributed a unique DOI. Please see: <https://royalsociety.org/journals/authors/author-guidelines/>

4) Data-Sharing and data citation

It is a condition of publication that data supporting your paper are made available. Data should be made available either in the electronic supplementary material or through an appropriate repository. Details of how to access data should be included in your paper. Please see <https://royalsociety.org/journals/ethics-policies/data-sharing-mining/> for more details.

<http://datadryad.org/submit?journalID=RSPB&manu=RSPB-2021-1204> which will take you to your unique entry in the Dryad repository.

Once again, thank you for submitting your manuscript to Proceedings B and I look forward to receiving your final version. If you have any questions at all, please do not hesitate to get in touch.

Sincerely,

Dr John Hutchinson, Editor

Associate Editor

Comments to Author:

I am impressed with the extent to which the authors have accepted the reviewer feedback from the first round of reviews. The study is undoubtedly compelling and adds important nuance to our understanding of the drivers of megafaunal extinction on Madagascar: The hypothesis of niche overlap between introduced livestock and native megafauna is compelling and this study represents an important initial step towards fully testing this hypothesis. My only concern is that this revision was only seen by a single referee, and not by the initial reviewers who recommended major revisions.

Reviewer(s)' Comments to Author:

Referee: 3

Comments to the Author(s).

I did not review the first version of this manuscript, but I have read through the author responses to reviewer comments from the first submission. The author responses seem to be thorough and clearly explained.

I learned much about the zooarcheology of Madagascar from reading this manuscript, and I thought the authors did a nice job establishing the context of their study. Since my expertise lies in bio(paleo)geochemistry, I focused on the presentation and analyses of these data. The authors apply the appropriate quantitative and modeling approaches (chronological and niche overlap) to the isotopic and radiocarbon data, and the results of these analyses support interpretations discussed in the paper. These are robust datasets that significantly expand the SI and 14C data from Madagascar during this time period.

Overall, the current version of this manuscript is well written and clearly presented - I do not have any substantive suggestions for edits or changes. The results are compelling, and fill in an interesting and important piece of the megafaunal extinction puzzle in Madagascar.

Decision letter (RSPB-2021-1204.R1)

28-Jun-2021

Dear Mr Hixon

I am pleased to inform you that your manuscript entitled "Late Holocene spread of pastoralism coincides with endemic megafaunal extinction on Madagascar" has been accepted for publication in Proceedings B.

Data Accessibility section

Open Access

Paper charges

Sincerely,

Appendix A

We thank the reviewers for their comments regarding early human arrival on Madagascar and attributes of the specimens used in this study.

Human arrival time:

- We agree with Reviewer 1 that the timing of human arrival relative to the extinctions gives important context to our work. To Reviewer 2's point, we choose not to review justification for early vs. late human arrival on Madagascar, because this is not the primary focus of our work and we provide reference to this ongoing debate (lines 58-63). For the purposes of our paper, and based on reference to a recent critical review of radiocarbon data associated with traces of human activity (Douglass et al. 2019 *Quat. Sci. Rev.*), we explicitly "assume that people were on the island 2,000-1,600 years ago" (lines 61-62). Though this estimate is relatively conservative, it does point to at least 500 years of coexistence between humans and megafauna before the megafaunal population crashes~1,000 years ago, and this lag is what we seek to explain. Additional lines of research are required to more confidently identify the earliest traces of human activity on the island, and this is beyond the scope of the paper.

Sample size limitations:

- We acknowledge the point of Reviewer 1 that additional ^{14}C and stable isotope data would strengthen the confidence of our conclusions regarding the potential for direct competition between ovicaprids and endemic megafauna. For this study, we sampled all known collections of ovicaprid bones from the region (housed in Madagascar, the U.S., and Australia), and additional sampling in this case is possible only through future fieldwork. Unfortunately, early sites in inland SW Madagascar that include an abundance of ovicaprid remains (e.g., Rezoky and Andranosoa, excavated mostly during the 1970s and 1980s) are currently unsafe to visit due to banditry. It is also worth noting that the poor preservation of bone proteins in the tropics contributes to sample size limitations.
- We modified the Discussion to highlight the fact that " ^{14}C datasets from goats and bushpigs are still limited" (lines 287-288). We also added a statement that "Future recovery and analysis of additional ovicaprid bones should be a priority, because even our limited dataset follows from sampling all known collections of ovicaprid bones from the region" (lines 334-336).
- Our existing data remain significant, because they are the first direct support for the ideas that ovicaprids and zebu cattle arrived in the region around the time of the disappearance of the megafauna and that potential for direct competition existed only between goats and certain endemic megaherbivores.

Specimen details and other points raised by Reviewer 2:

- Line 51 - Cattle present on Madagascar have been variously referred to as *Bos indicus*, *Bos taurus*, and *Bos madagascariensis*, but these species have been reclassified to *Bos taurus*. Zebu cattle (of the subspecies *Bos taurus indicus*) colonized SW Madagascar, so we have changed all references to "*Bos indicus/taurus*" to "*Bos taurus indicus*."
- Lines 54-56 – We appreciate this comment on the organization of our Introduction. Most authors agreed that it is best to present our approach early in the Introduction and to elaborate on it in the following paragraphs.
- Lines 69-71 – We reference the sparse terrestrial Pleistocene paleontological record from Madagascar, because this relates to the question of how extinct megafauna responded to

past environmental change (the topic of the previous sentence). Unfortunately, we have limited traces of these responses before the mid-Holocene.

- Line 89 – Indirect and direct forms of competition are part of the “subsistence shift hypothesis” as we have defined it (with reference to Godfrey et al. 2019 *J. Hum. Evol.*). We do not believe that we misrepresent this hypothesis as it was stated in 2019, for one of us was a co-author of the paper that introduced the “subsistence shift hypothesis,” and L. Godfrey kindly provided positive comments on our manuscript.
- Line 89 onwards – We acknowledge that apparent competition (like other forms of competition) is difficult to identify in nature, which is why we seek to evaluate merely the potential for different types of past competition. The idea that agriculture and pastoralism facilitated the growth of populations of human hunters at the expense of endemic megafauna is at the heart of the “subsistence shift hypothesis,” and we refer to this as “apparent competition” for clarity. We are happy to consider other suggested terms that describe this hypothetical situation.
- Line 100 – The question on the length of temporal overlap is a good one that we have discussed with researchers not involved in the study. In this paper, we begin by focusing on addressing the basic question of whether there was any temporal overlap and potential for interaction. Future research may clarify differences in the length of temporal overlap between introduced and endemic megaherbivores in other parts of the island. Note also that we modified the text (line 101) to make it clear that we focus here on testing the “subsistence shift hypothesis.” The “synergy hypothesis” is a multifaceted one that is given for context and to present a plausible alternative.
- Line 157-158 – We reviewed data from the two extinct lemurs for the reasons that we specify on lines 159-161 (relatively large pre-existing ¹⁴C dataset in one case and semi-terrestrial locomotion in the other). It is unlikely that this choice introduces biases, because we do not make generalizations about other extinct lemurs based on the two extinct lemurs that we chose to consider. The common name “monkey lemur” is frequently used in reference to *Archaeolemur majori*, and we have deleted “giant” from the subsequent sentence (line 159).
- Line 159 – We modified the wording on line 162 to clarify the fact that we excluded 39 specimens from the “combined data” (as opposed to from the two lemurs in particular). Details the decisions behind this exclusion can be found in the Appendix “Additional Data” section. Questions on taxonomic assignments and taxon-specific sample sizes for the various datasets are reasonable, and they are answered in multiple pages of text in the Appendix. Specifically, lines 154-155 refer to details in “Appendix ‘Sample Collection’ & ‘Laboratory Analyses,’” and line 157 refers to details in “Appendix ‘Additional Data.’” We agree that additional work to refine the taxonomic resolution of this study could reveal interesting patterns of niche partitioning and asynchronous extinction that are currently cryptic.
- Line 169 – We reworded this sentence to clarify that we use GLMs to compare “influences of taxon, space, and time on stable isotope values,” and we added a reference to “Appendix ‘Data Analysis,’” which, along with Tables S2 & S3, further clarifies how we used the GLMs.
- Line 205 – We chose to highlight the maximum temporal overlap (between zebu and *P. insignis*), but we also reference Appendix Table S1 (line 204), which includes all confidence and credible intervals for reference.

- Line 279 – Discussion, opening paragraph – We focus the Discussion on our test of the “subsistence shift hypothesis,” because the “synergy hypothesis” is presented as a plausible alternative. However, in the paragraph between lines 338 and 354, we do discuss how our data reveal differences between how introduced livestock and endemic megaherbivores likely responded to past forest clearance and drought (related to the “synergy hypothesis”). Our data give a critical test of the “subsistence shift hypothesis,” but they do not permit us to falsify the “synergy hypothesis.”
- Competition, drought, and habitat clearance ~1000 years ago all could have contributed to the demise of the megafauna, and only future research from multiple sites with refined spatio-temporal resolution can help us “tease out the relative effects of competition versus habitat loss.” The main contribution of our research follows from our test for the potential for past competition. We cannot exclude the possibility that drought contributed to the extinctions, but we do reference past work on this topic (lines 65-69).
- Lines 297-8, Lines 307-8 – These statements in isolation may “feel like unsupported assumptions,” but they contribute to a plausible explanation for why the disappearance of a diverse array of megafauna likely coincided with the arrival of a suite of introduced livestock. Other explanations could be given (e.g., disease), but they appear less substantiated based on existing data.
- Lines 329-330 – Our suggestion that “people may have encouraged ovicaprids to forage on a wide range of plant types in deference to the grazing preferred by zebu” is not essential to our argument and is based on modern observations of the authors, so we modified the wording of this sentence accordingly.
- Line 322 – We are unfortunately missing vegetation reconstructions from inland SW Madagascar, and we agree that it would be interesting to compare regional differences in herbivore diet against records of past vegetation change. Currently, the spiny thicket ecoregion is defined by similar vegetation that exists at both the coastal and inland sites that we consider. “Coastal” is defined in the Appendix as <10 km from shore, and we added this detail to lines 142-143 for clarity.
- Line 330-331 – We support this statement with a reference, but we reworded the beginning of this sentence to clarify that “The dispersal ability of giant tortoises may attest to high tolerance of resource depression.”
- Line 337 – The statement that “aridification could have been a greater stress for introduced livestock” is part of the topic of the paragraph, and the reasoning follows. We have clarified this by prefacing the sentence with “For the reasons that follow,” As noted on lines 353-354, “consistent past reliance on relatively moist habitat by ancient zebu [as evidenced by relatively low $\delta^{15}\text{N}$ values] may reflect their sensitivity to drought.”
- Line 343 – We write that forest clearance in the study region “may have been anthropogenic,” because there is debate on how extensively past humans cleared forests on Madagascar. Some declines of arboreal taxa ~1,000 years ago in coastal SW Madagascar have been attributed to drought and marine surges. For an example, see Virah-Sawmy et al. (2010, *J of Biogeog.*). The causes of past vegetation change are mostly beyond the scope of this paper, but we do suggest that the loss of endemic megaherbivores and spread of pastoralism contributed to past vegetation change (lines 361-370).
- Line 346 – We acknowledge that we have inconsistent use of “tracking” in the manuscript, because we state that some animals “track” by moving (line 305) and that

others “track” by persisting in place (line 349). We have modified the wording accordingly and so that “tracking” consistently refers to movement.

- Paragraph starting line 353 – We agree that we have not considered all megafaunal taxa, so we preface this statement with “In our existing sample,” (lines 356-357). We later cite another study to suggest that Madagascar lacked an endemic grazer guild (line 360). This paragraph is meant to put our results in the context of the work of others and to summarize lasting consequences of the extinctions that we discuss. We do think that this topic is relevant to our results from introduced herbivores, because we find little evidence to suggest that introduced grazers are fulfilling the same roles as extinct endemic megaherbivores.
- Line 372 – We agree that habitat loss in the form of deforestation could have contributed to megafaunal extinction. However, this is not the focus of our research, and we do discuss both how livestock could have contributed to deforestation (lines 304-306 and lines 369-370) and how endemic megafauna could have been relatively sensitive to changes in plant communities (lines 344-346).
- Figures – We changed the earlier reference of *Vorombe* to *Aepyornis* for clarity (line 50), and all details on our treatment of taxonomic units are given in the Appendix, as noted previously. Most *Aepyornis* data come from previous work with eggshell, and, as noted in the Appendix, we identified *Aepyornis* only “to the genus level because eggshell is difficult to assign to species. Although the majority of skeletal material belongs to *A. maximus* in southwestern Madagascar, *A. hildebrandti* is not unknown (Hansford & Turvey 2018).”